# Woodland Cover Change in the Central Rift Valley of Ethiopia

**Demamu Mesfin [1],***, **Belay Simane [2]**, **Abrham Belay [3]**, **John W. Recha [4]** and **Habitamu Taddese [5]**

[1] Department of Environmental Science, College of Forestry and Natural Resources, Hawassa University, P.O. Box 05, Hawassa, Ethiopia

[2] Center for Environment and Development, College of Development Studies, Addis Ababa University, P.O. Box 56649 Addis Ababa, Ethiopia; belay.simane@aau.edu.et

[3] Department of Natural Resource Economics and Policy, College of Forestry and Natural Resources, Hawassa University, P.O. Box 05, Hawassa, Ethiopia; abrish.z2010@gmail.com

[4] CGIAR Research Program on Climate Change, Agriculture, and Food Security (CCAFS), International Livestock Research Institute (ILRI), P.O. Box 30709-00100 Nairobi, Kenya; j.recha@cgiar.org

[5] Department of Geographic Information Science (GIS), College of Forestry and Natural Resources, Hawassa University, P.O. Box 05, Hawassa, Ethiopia; habtu1976@gmail.com

* Correspondence: demmesfin@gmail.com; Tel.: +251-911704421

**Abstract:** Woodlands, which are part of the landscape and an important source of livelihood for smallholders living in the environmentally vulnerable Central Rift Valley (CRV) of Ethiopia, are experiencing rapid changes. Detecting and monitoring these changes is essential for better management of the resources and the benefits they provide to people. The study used a combination of both quantitative and qualitative methods to analyze the extent and pattern of woodland cover changes from 1973 to 2013. Pixel-based supervised image classification with maximum likelihood classification algorithm was used for land cover classification and change detection analyses. Local peoples' perceptions were used to explain the patterns of change and their possible reasons. Four major land cover classes were identified, with an overall accuracy of 88.3% and a Kappa statistic of 0.81 for the latest image. The analysis revealed a major land cover reversal, where woodland (92.4%) was the dominant land cover in 1973, while it was agriculture (44.7%) in 2013. A rapid reduction in woodland (54%) and forest (99%) covers took place between 1973 and 2013, with the majority of the conversions being made during the government transition period (1973 to 1986). Agriculture (3878%) and grassland (11,117%) increased tremendously during the 40-year period at the expense of woodlands and forests. Bare land increased moderately (40%). Thus, woodlands are under increasing pressure from other land uses, particularly agriculture, and declining faster. If the current trends of land cover change remain unabated it is likely that woodlands will disappear from the landscape of the area in the near future. Therefore, better forest policy and implementation tools, as well as better woodland management strategies and practices, need to be in place for woodlands to continue providing vital ecosystem goods and services to the local people, as well as to the environment.

**Keywords:** woodland; land cover change; pattern of land cover change; central rift valley

## 1. Introduction

Woodland is land spanning more than 0.5 hectares with trees higher than 5 m and a canopy cover of 5–10 percent, or trees able to reach these thresholds in situ; or with a combined cover of shrubs, bushes and trees above 10 percent [1]. They provide a wide range of economic, social, and ecological benefits, ranging from cultural to tangible economic values [2,3]. Under proper

stewardship, this important capital asset can play a critical role in human livelihood, as well as in ecosystem functioning and health [4]. They are also important for helping people adapt to the impacts of climate change [5–7]. These wide-ranging roles of woodlands have received renewed recognition in recent times [8].

However, these important resources are experiencing changes. One of the most important changes that the world is experiencing today is land-cover change [3,9–12]. Of all land cover changes, forest/woodland cover change or deforestation is one of the oldest and most important changes that people have made to the surface of the earth [3,13,14]. Woodlands are still under threat by various extractive human activities that result in degradation of habitat, land fragmentation, and land-use change (e.g., from forestry to agriculture) [2]. Climate change can also exacerbate these pressures and exert significant negative impacts on the capacity of woodlands to provide vital ecosystem services [15].

As a result of immense human pressure, the world's forest and woodland covers have been greatly reduced. They once covered about six billion hectares of the earth's land surface several years ago, while they are currently estimated at just over four billion hectares, about 31% of the land surface [1]. The same report indicates that net deforestation at the global level occurred at a rate of 0.14 percent per year between 2005 and 2010. During the same period, Africa lost 3.4 million hectares of forest annually. This makes Africa one of the largest net losers of forests in the world, next to South America, which has an estimated loss of around 4 million hectares per year.

Forest clearance in Africa is driven mainly by the demand for land for cultivating crops and grazing, and for fuel [3]. The heaviest forest losses in sub-Saharan Africa happen in areas where wood is needed for fuel or where forestland is needed for growing crops [1,3,16]. Ethiopia is one of the countries in sub-Saharan Africa experiencing such challenges.

Ethiopia, with a land area of over 1.12 million km$^2$ and a population of over 100 million people, is experiencing a huge deforestation rate [17–21] and land cover change [22–24]. For example, the rate of deforestation in Ethiopia between 2005 and 2010 was reported to be 1.11% per year, which is considerably higher than the world's 0.14% and Africa's 0.5% during the same period [1].

Currently, Ethiopia's forest cover is reported to be 11.4% by the FAO [25], and 15.5% by the MEFCC [26]. The FAO [1] estimate (11%), which was derived from the report of Woody Biomass Inventory and Strategic Planning Project WBISPP [20], was also very close to the current FAO estimate. In the FAO [1] report, high forests (i.e., natural forests with tall trees), plantations, and high woodlands were reclassified as 'forest', while low woodlands and shrublands were reclassified as 'other wooded land'. In the recent reports of both MEFCC [26] and FAO [25], Bamboo forests and dense high woodlands are included as forests.

According to WBISPP [20], Ethiopia has just over 4 million hectares of high forest (3.56% of the area of the country), 29.24 million hectares of woodland (25.5%), and 26.4 million hectares of shrubland (23.1%). Acacia woodlands cover about 55% of the total woodland area, i.e., more than the other types of woodlands altogether [27]. Acacia woodlands are mainly concentrated in the lowlands of the Rift Valley [27,28] and are the climax vegetation of the area [20].

Indeed, Acacia woodlands have been one of the characteristic features of the landscape of particularly the CRV and important source of livelihood for the people living in and around the area. They provide a wide range of goods and services for the local people and the nation at large. For example, the majority of charcoal that is being used by the citizens of Addis Ababa, Ziway, Shashemene, and Adama towns is coming from the CRV woodland [27,29].

For the rural people living in the CRV, the Acacia woodlands are used to obtain alternative means of livelihood [30]. They are also a major source of income and a contingency food particularly during crop failures which are not uncommon due to shortage and uneven distribution of rainfall, drought, high rate of evapotranspiration and shortage of rivers and streams which can be used for irrigation [23,31]. Moreover, woodlands can play an important role in improving the adaptive capacity of the local people [16]. These socio-economic contributions of woodlands are in addition to their ecological roles in safeguarding the fragile ecosystem of the CRV area [27,32].

Several studies have been conducted in Ethiopia to examine forest/woodland decline and land-use and land-cover changes. However, very few have attempted to address land-use and land-cover issues in the center of the Main Ethiopian Rift (MER) [17,23,31,33–35]. Most of these studies were either carried out far away from the main CRV, or were focused on overall land changes, rather than woodland cover changes. The later entails the need to have woodlands at least in the past.

Moreover, studies that examine woodland cover change across the different governance regimes, where there was marked land administration differences, are insufficient. During the Imperial period (pre-1974) land was owned by the state and used by few landlords. The socialist regime succeeded the Imperial regime in 1974 and declared "land for the tiller" in 1975, where land was put under the ownership of the public and tenants became land tillers. From 1991 onwards, after the overthrow of the socialist regime, land was declared as both a state and public property, meaning that farmers only had user rights. The association of these governance regimes and the transitional periods on the status of woodland resources was not adequately addressed [31].

To get a better view of woodland changes in the main CRV area this study included all areas between the Three Lakes (Langano, Abijata, and Shalla). Hence, this paper tries to answer the following questions: (1) how have the woodland resources in the CRV changed over time? and (2) what do the patterns of change look like?

Several methods have been used to detect land cover change using remotely sensed data. A review of the latest methods can be found in Halmy, Gessler [36]. Satellite imagery is one of the most commonly used sources of information and analysis for land use and land cover change [37]. This study followed remote sensing techniques to detect woodland cover change in the CRV area and used qualitative data obtained through focus group discussions (FGDs) and key informant interviews (KIIs) to explain the reasons for the change.

## 2. Materials and Methods

### 2.1. The Study Area

This study was conducted in the Central Rift Valley of Ethiopia (7°22′–7°42′ N, 38°25′–38°47′ E) (Figure 1), about 200 km south of the capital, Addis Ababa. It is bounded by three lakes: Lake Langano in the East, Lake Abijata in the West, and Lake Shalla in the South and Southwest. The altitude varies from 1560 m a. s. l. at the Lake Shalla shoreline to 2061 m a.s.l. at the peak of Mount Fiqe, which is found between Lake Abijata and Lake Shalla.

The natural vegetation of the study area can generally be characterized as open Acacia woodland albeit patches of dense woodlands and forests can also be found. The most common species along the rift floor are deciduous Acacia, while a wide variety of both deciduous and coniferous trees are found on higher elevations and areas with better soil and moisture conditions. Acacia species available in the woodland include *Acacia tortilis*, *Acacia senegal*, *Acacia seyal,* and *Acacia etbaica*. These tree species, particularly the first two, are the dominant Acacia species in the CRV area [27]. Other tree species found in the study area include *Balanites aegyptiaca*, *Maytenus senegalensis*, *Dichrostachys cinerea*, *Euphorbia candelabrum*, *Ziziphus mucronata*, *Croton macrostachys*, *Cordia africana* and *Ficus vasta*.

In addition, several semi-evergreen shrubs, such as Carissa edulis, Croton dichogamous, Solanum schimperianum, Sclerocarry birrea, Terminalia browni, Harisonia abyssinica, Capparis tomentosa, and Acokanthera schimperi, are found especially on rocky and degraded sites either scattered over the study area or mixed with the dominant Acacia trees [27,31,38]. The lowest vegetation level of the woodlands is usually covered by different species of grasses, such as Cenchrus, Cynodon, Dactyloctenium, Digitaria and Sporobolus [38], which are important for both livestock and wild animals. In most parts of the study area, the stocking and species diversity of the woodland appears to be severely affected due to human interference.

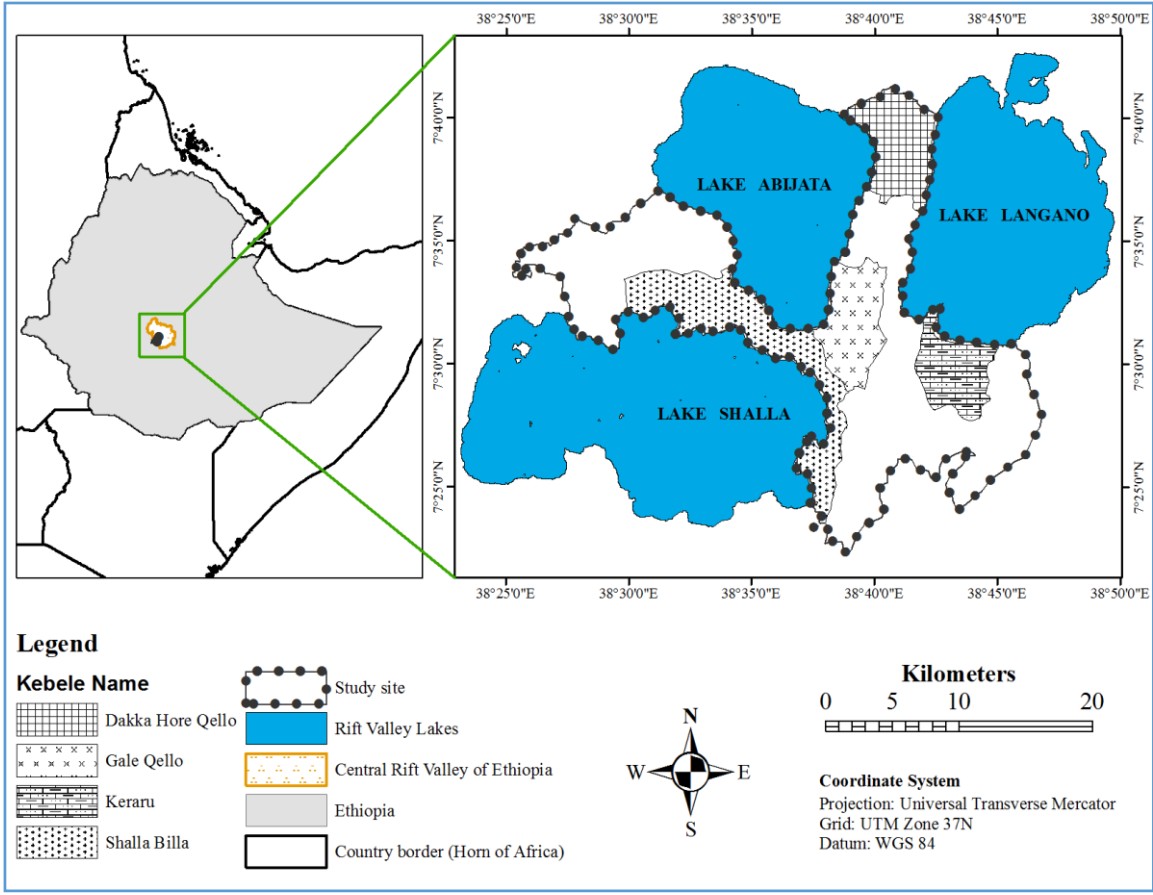

**Figure 1.** Map of the study area.

The study site was purposely selected for the following reasons: (1) it was known for its dense Acacia woodland [27], to the extent that it was not possible to see the three Lakes while driving along the main road that passes through them just a couple of decades ago (personal observation), and hence the people who are dependent on woodlands for goods and services will be more vulnerable to woodland changes both socially and economically [5]; (2) it is one of the most environmentally fragile areas in Ethiopia, where small changes in natural resources can have far reaching consequences on ecosystem goods and services, and potentially undermine the role they play for society and the environment [32,33]; (3) it is an area typical of a rapidly changing landscape in Ethiopia, if not in Africa [39]; (4) there is a national program to, among other thing, establish an Acacia Woodland Reserve and Community-Based Carbon Sequestration Project in the Ethiopian Rift Valley System as national adaptation options [40], and hence this study can provide some information about the Acacia woodlands of the CRV and their likely contributions, particularly to the local people and the environment.

*2.2. Data Collection and Analysis*

The study followed a mixed research methods approach for collecting both quantitative and qualitative data needed to address the research problem. The core assumption for using the approach is that it provides a more complete understanding of the research problem than either approach alone [41]. Moreover, this study adopted a convergent parallel mixed methods design, where both quantitative and qualitative data are collected at about the same time and converged to provide a comprehensive analysis of the problem.

Remote sensing techniques and field observations were used to collect the woodland cover change data and to analyze the extent and pattern of the change over the last forty years (1973–2013).

Landsat images were used in this research because of the rich archival data availability that meets our requirement for temporal variation. The images used in this study were selected based on the criteria that they were captured during the dry season when the vegetation contrasts with the other land cover types and cloud cover in the images was below 10%. Key informant interviews (KIIs) and focus group discussions (FGDs) were also conducted to examine people's perception of woodland cover change and to find explanations for the change.

### 2.2.1. Land Use/Land Cover Data Acquisition and Analysis

Landsat images of different time stamps were used for analyzing the land use/cover change in the study area. A total of six images representing six different time points were used to study the changes over the last forty years (1973–2013) and they were obtained from three different Landsat sensors: Landsat MSS (Multi-Spectral Scanner), Landsat TM (Thematic Mapper), and Landsat ETM+ (Enhanced Thematic Mapper plus). These time points were mainly selected to reflect the changes that the Acacia woodlands in the central rift valley have experienced over the three different regimes of Ethiopia, namely, the Imperial period pre-1974, the Derg or 'socialist' period between 1974 and 1991, and the current regime since 1991. Due to the limited availability and quality of satellite images, one satellite image was analyzed both for the imperial and socialist periods each, while it was possible to analyze four images for the current regime. Accordingly, the land-use/cover changes between the six periods (i.e., 1973, 1986, 2000, 2005, 2011, and 2013) were quantified and a change detection matrix for all periods was derived. The detail of the data used and the system characteristics of their sources is given in Table 1.

**Table 1.** Characteristics of the remotely sensed datasets used for land cover analysis in this study.

| No. | Image Data Type (Path & Row) | Acquisition Date | Type of Sensor | Spatial Resolution |
|-----|------------------------------|------------------|----------------|--------------------|
| 1 | Landsat MSS (p168r55) | 1 January 1973 | MSS | 57 m * 79 m for all bands |
| 2 | Landsat TM (p168r55) | 21 January 1986 | TM | 30 m multispectral, 120 m thermal |
| 3 | Landsat ETM+ (p168r55) | 5 February 2000 | ETM+ | 15 m panchromatic, 30 m reflective, 60 m thermal |
| 4 | Landsat ETM+ (p168r55) | 12 December 2005 | ETM+ | 15 m panchromatic, 30 m reflective, 60 m thermal |
| 5 | Landsat ETM+ (p168r55) | 1 January 2011 | ETM+ | 15 m panchromatic, 30 m reflective, 60 m thermal |
| 6 | Landsat ETM+ (p168r55) | 12 January 2013 | ETM+ | 15 m panchromatic, 30 m reflective, 60 m thermal |

The images were corrected for geometric and atmospheric errors using ERDAS imagine 9.1 software. The geospatial data used in this study have spatial reference systems defined within WGS84, Zone 37N grid of the Universal Transverse Mercator (UTM) projection. Pixel-based supervised image classification with maximum likelihood classification algorithm was employed for land cover classification. The near-infrared, red and green spectral bands of the images were combined to visualize the data for collecting training areas used for classification. High-resolution images in Google Earth and ground truth data were also used to support delineating the training areas on the recent (2013) satellite image. A total of 179 ground-truthing points collected from the field were used for validating the results of the 2013 image classification. Then, the land cover change detection was implemented by making a pairwise comparison of the time series land cover data. ArcGIS 9.3 software was used for mapping the land cover data.

### 2.2.2. Characteristics of Land Cover Types

Six different land categories were identified as a primary unit for characterizing the land cover of the study area. The descriptions used in the classification of the land cover types are presented in Table 2. The secondary units identified and quantified are presented in Section 3.2 They include the area of each land cover type in hectares, land cover change in hectares and change trajectories, and observational products such as the land cover maps and the radar chart output.

**Table 2.** Description of land cover types identified in the CRV of Ethiopia.

| No. | Land Cover Type | Description |
|-----|-----------------|-------------|
| 1 | Forest | Land covered by vegetative communities comprised principally trees of the same or different species. This class represents woodlots around home gardens, patches of remnant natural forests on slope sides and stretches of riverine forests, forests found near rivers and lakes. Woodlots around homesteads are mainly used as a source of wood, windbreaks, and live fences, whereas the remnant natural forest is mainly used for livestock grazing and browsing just like woodlands. |
| 2 | Woodland | Land covered predominantly by Acacia trees. Some species of trees, shrubs, and bushes may also be found mixed with the Acacia trees. Different species of grasses can normally be found under the canopies of the Acacia woodland and in open spaces. This cover type dominates the area between the three lakes (Shalla, Langano, and Abijata) where Acacia is the dominant species. This land cover class is mainly used for grazing and browsing by both domestic and wild animals and is also the source of wood for households. |
| 3 | Grassland | This is a landscape largely dominated by grasses and other occasional herbaceous plant communities separated by intervening bare space. A very small proportion of trees and shrubs may exist in this category. This cover type prevails around the lakes, particularly Lake Abijata, marking the continuous lake secession process in which the transition from bare land when the lake retreats is followed by a succession of grass communities. They are predominantly used for grazing. |
| 4 | Cropland | Land allotted to the cultivation of mainly annual food crops. This land cover is found across all parts of the study area and is predominantly a rain-fed type of production system. |
| 5 | Bare land | A cover class represented by a rock outcropping, roads, eroded surface, settlement areas, silt deposition, and lake secession sites that are largely devoid of above-ground vegetation. |
| 6 | Others | This category represents small areas that are difficult to classify as one of the above land cover classes. They may refer to shadowed areas, infrastructures, and a mixture of two or more land cover categories per pixel. |

Source: Authors.

### 2.2.3. Focus Group Discussion

Four Focus Group Discussions (FGDs) representing each selected Kebele were held to assess the perceptions of the local people towards woodland cover change and to find the possible reasons for the change. The main questions raised for discussion were: (1) have you noticed woodland cover change in the area; (2) if yes, how do you perceive the changes; (3) how do you explain the patterns of the change; and (4) what are the reasons for the change. The group participants were people who know better about the overall condition of the area. The group mainly consists of elderly people albeit women and youth were also represented. The group size was limited to 6 to 10 people and the total number of people participated in the discussion was 31. The discussions were held under a tree canopy in their vicinities, which they are comfortable with, and facilitated by a moderator, the researcher himself, a translator, and a note-taker. Points were noted when the consensus among group members was reached. The qualitative data captured in this manner were then narrated to provide explanations for woodland cover change and to supplement the quantitative data obtained from image analysis. Additionally, the development agents were always around the group for any assistance needed, including making prior communications and appointments with the local people.

### 2.2.4. Key Informant Interview

Key informant interviews (KII) were conducted for specialized information particularly concerning the condition of the woodland in the area. The key informants include district experts, Abijata-Shalla

Lakes National Park chief warder and experts, development agents, and Kebele council members. A total of 13 key informants were interviewed. These individuals were selected because they are supposed to have special information and knowledge which others do not possess (e.g., forest policies, institutional changes, ownership issues, forest practices, and actions taken on culprits regarding woodland use). A semi-structured checklist with interactive conversions was made with the informants. This interview was conducted to enrich and substantiate the outcomes of FGDs.

## 3. Results and Discussion

### 3.1. Accuracy Assessmet

The overall classification accuracy for the latest image was 88.3% with the Kappa statistic of 0.81 (Table 3). The Kappa statistic result shows a high level of agreement for the classified image. User's and producer's accuracies of individual classes ranged from 81.8% (for grassland) to 91.7% (for bare land), and 73.3% (for bare land) to 95.6% (for woodland), respectively. As confirmed by the ground-truthing work, there were errors during the classification of the image into different land cover classes. However, the error between the classes was low. The relatively higher confusions occurred between woodlands and agriculture and agriculture and bare land. For example, ten out of the 75 randomly generated samples of woodlands were incorrectly classified as agricultural land, while two out of 81 agricultural lands were incorrectly classified as woodlands. Similarly, four out of 81 agricultural lands were incorrectly classified as bare land, while there was no bare land incorrectly classified as agricultural land. These errors are acceptable as the overall accuracy for effective land use and land cover change analysis is higher than the 85% minimum threshold level [42,43].

**Table 3.** Error matrix of classification accuracies for 2013.

| Classification | Woodland | Agriculture | Grassland | Bare land | Ground truth points | User's Accuracy, % |
|---|---|---|---|---|---|---|
| Woodland | 65 | 10 | 0 | 0 | 75 | 86.7 |
| Agriculture | 2 | 73 | 2 | 4 | 81 | 90.1 |
| Grass land | 1 | 1 | 9 | 0 | 11 | 81.8 |
| Bare land | 0 | 1 | 0 | 11 | 12 | 91.7 |
| Total | 68 | 85 | 11 | 15 | 179 | |
| Producer's Accuracy, % | 95.6 | 85.9 | 81.8 | 73.3 | | |

Overall accuracy = 88.3%; Kappa statistic = 0.81. *Source:* Authors.

### 3.2. State of Woodland Cover during 1973–2013

The woodland cover change and the rate of change between the reference periods in the study area over the last forty years (1973–2013) are presented in Tables 4 and 5, respectively. Figure 2 shows the spatial distribution of the different land cover types in the reference years chosen, i.e., in 1973, 1986, 2000, 2005, 2011, and 2013, and Figure 3 displays shifts in the pattern of land cover types for the entire study period (1973–2013). The Radar Chart shows a unidirectional change mainly from woodland to agriculture and grassland.

Ninety-five percent of the total land area in 1973 was covered by tree-dominated vegetation, like woodlands (92.4%) and forests (2.7%), while bare land and agriculture accounted for less than 5% of the area (Table 4). However, this pattern of land cover has dramatically changed during the study period. Woodland cover dropped from 92.4% in 1973 to 69.7%, 56.1%, 50.1%, 50.5%, and 42.4% in 1986, 2000, 2005, 2011, and 2013, respectively (Table 4). In 2013, in just forty years, a major reversal in land cover has occurred; agriculture being the dominating land cover type (44.7%). Grassland also expanded from almost nothing to 8%. Hence, agriculture and grassland have significantly expanded at the expense of woodlands and forests.

In addition to the land cover changes that the area has experienced, the rate of change between the periods is even more worrisome. For example, agriculture increased by 3878.2% over the last forty years, i.e., it increased from only 467 ha in 1973 to 18,578 ha in 2013. Likewise, grassland increased by 11,116.7%, i.e., from almost nothing to 3365 ha, over the same period. On the other hand, there was a loss of 54.1% and 99.1% of woodland and forest cover between 1973 and 2013, respectively. This indicates that there have been drastic changes in land use and land cover of the area during the study period.

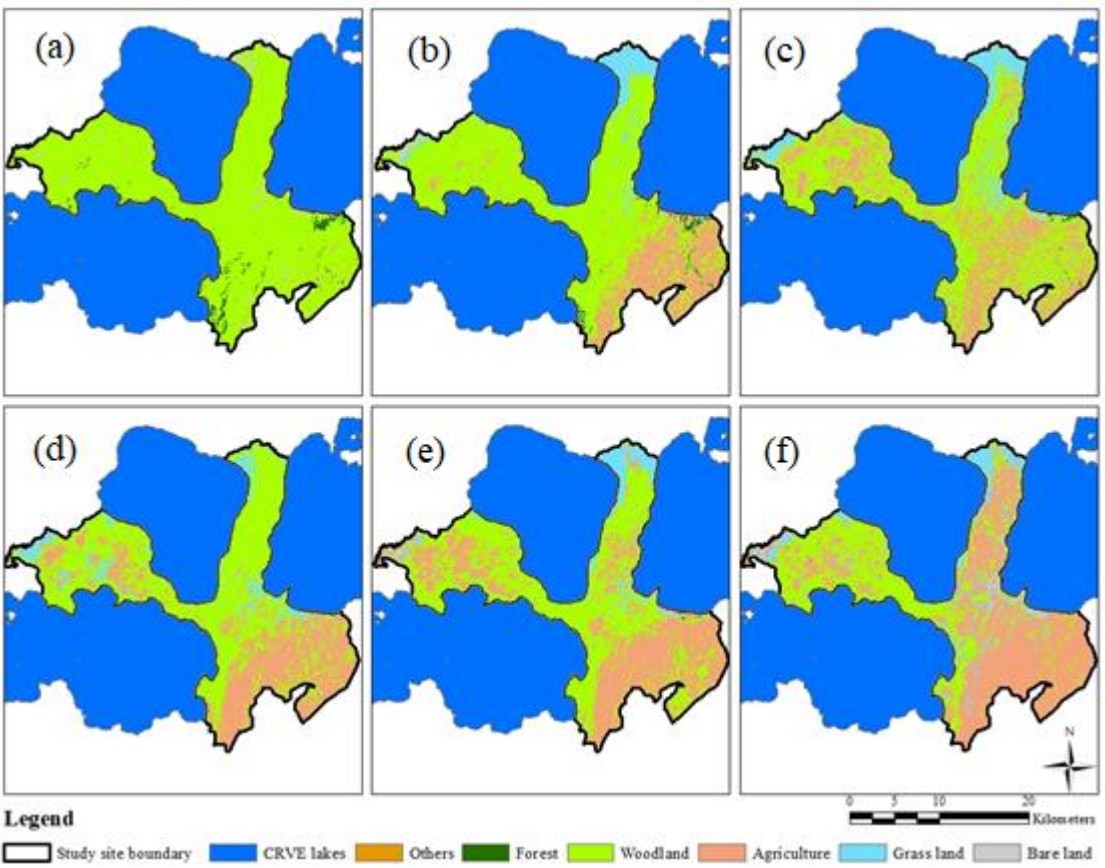

**Figure 2.** Land cover of the study area in (**a**) 1973, (**b**) 1986, (**c**) 2000, (**d**) 2005, (**e**) 2011 and (**f**) 2013.

**Table 4.** Area in hectare and the percentage of the land cover of each class in each reference year.

| Cover Types | 1973 | | 1986 | | 2000 | | 2005 | | 2011 | | 2013 | |
|---|---|---|---|---|---|---|---|---|---|---|---|---|
| | Ha | % | Ha | % | Ha | % | Ha | % | Ha | % | Ha | % |
| Woodland | 38417 | 92.4 | 28022 | 67.4 | 23328 | 56.1 | 20842 | 50.1 | 21002 | 50.5 | 17633 | 42.4 |
| Forest | 1141 | 2.7 | 562 | 1.4 | 221 | 0.5 | 10 | 0.0 | 79 | 0.2 | 10 | 0.0 |
| Agriculture | 467 | 1.1 | 8790 | 21.1 | 12181 | 29.3 | 16792 | 40.4 | 16323 | 39.2 | 18578 | 44.7 |
| Grassland | 30 | 0.1 | 3555 | 8.5 | 5077 | 12.2 | 3747 | 9.0 | 3463 | 8.3 | 3365 | 8.1 |
| Bare land | 1351 | 3.2 | 600 | 1.4 | 659 | 1.6 | 122 | 0.3 | 529 | 1.3 | 1891 | 4.5 |
| Others | 191 | 0.5 | 68 | 0.2 | 131 | 0.3 | 84 | 0.2 | 200 | 0.5 | 120 | 0.3 |

**Table 5.** Land cover change in hectare and percentages of the change for each land cover class between the periods indicated.

| LC Types | 1973–1986 | | 1986–2000 | | 2000–2005 | | 2005–2011 | | 2011–2013 | | 1973–2013 | |
|---|---|---|---|---|---|---|---|---|---|---|---|---|
| | Ha | % | Ha | % | Ha | % | Ha | % | Ha | % | Ha | % |
| Woodland | −10395 | −27.1 | −4694 | −16.8 | −2486 | −10.7 | 160 | 0.8 | −3369 | −16.0 | −20784 | −54.1 |
| Forest | −579 | −50.7 | −341 | −60.7 | −211 | −95.5 | 69 | 690.0 | −69 | −87.3 | −1131 | −99.1 |
| Agriculture | 8323 | 1782.0 | 3391 | 38.6 | 4611 | 37.9 | −469 | −2.8 | 2255 | 13.8 | 18111 | 3878.2 |
| Grassland | 3525 | 11750.0 | 1522 | 42.8 | −1330 | −26.2 | −284 | −7.6 | −98 | −2.8 | 3335 | 11116.7 |
| Bare land | −751 | −55.6 | 59 | 9.8 | −537 | −81.5 | 407 | 333.6 | 1362 | 257.5 | 540 | 40.0 |
| Others | −123 | −64.4 | 63 | 92.6 | −47 | −35.9 | 116 | 138.1 | −80 | −40.0 | −71 | −37.2 |

Source: Authors.

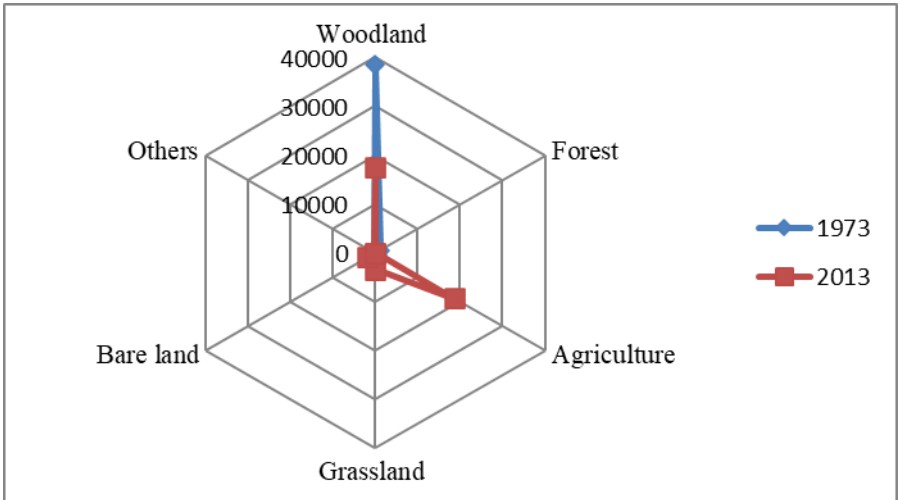

**Figure 3.** Radar chart displaying shifts in the pattern of land cover change in hectares between 1973 and 2013.

### 3.3. Land Use/Land Cover Changes in the Main Central Rift Valley Area

3.3.1. General Trends of Land Use and Land Cover Change

The findings of this study demonstrate the dynamics of land use and land cover changes in the main CRV of Ethiopia over the last four decades on both spatial and temporal scales. These changes were mainly associated with woodland cover change. Up until 1973, the study area was predominantly covered by woodland. However, it has experienced substantial land use and land cover changes since then, and these changes have led to marked differences in vegetation cover and land-use patterns over the study period (1973 to 2013). Currently, four major land cover types are found, namely, woodland, agriculture, grassland, and bare land. Forests have virtually disappeared. The major transition in land cover change took place between woodland and agriculture and grassland. As a result, the woodland cover has substantially declined, while agriculture and grassland covers have enormously increased.

Several studies conducted in the CRV area have also reported similar results in suggesting that there has been a loss of woodland cover and that it has been converted mainly to agriculture and grassland [23,31,33,34]. Biazin [31] reported a reduction in woodland cover (both dense and scattered trees) from 77% in 1965 to 40% in 2010. This result is close to the current finding. Garedew, Sandewall [23], on the other hand, reported a decline in woodland cover from 40% to 9% for Keraru and from 54% to nothing for Gubeta Arjo Kebeles between 1973 and 2000. Keraru was also part of this study, whereas Gubeta Arjo was not included, as it is a bit far from the Acacia-covered lowlands of the CRV, and most parts of the Kebele are also classified as Midlands, with different vegetation types.

According to Muzein [34], more than 37% of the total Abjata-Shalla Lakes National Park area lost its vegetation cover and agriculture was responsible for the loss of 80% of the total terrestrial productive ecosystem. Meshesha, Tsunekawa [33] also reported a 69% woodland loss in the CRV. Although the trend of land cover change was similar, the rate of change reported by these studies seems to vary, as their studies covered different sites and sizes of areas and slightly different periods as well. For example, Meshesha, Tsunekawa [33] covered 26 districts, including adjacent highlands, where agriculture is intensively practiced. Moreover, the definitions given to the different land cover types might have contributed to its share.

Several results indicating forest decline outside the main CRV of Ethiopia were reported. For example, Dessie and Kleman [17], Kindu, Schneider [44], and Abate and Lemenih [22] studied land use and land cover changes in the southern parts of the country and reported a total natural forest cover loss of 82%, 24.4%, and 22.64% between 1972 and 2000, 1973 and 2012, and 1973 and 2004, respectively. Kindu, Schneider [44] also reported 36% and 24.8% woodland cover and plantation forest losses, respectively. Studies of forest cover change in the eastern and western parts of the country are limited.

However, Tsegaye, Moe [35] reported a 97% reduction in woodland cover in the northern Afar rangelands in eastern Ethiopia between 1972 and 2007. Similarly, Emiru and Taye [45] reported almost a complete disappearance of natural forests and a massive reduction in woodland cover in the western part of Ethiopia between 1957 and 2006. Alemu, Garedew [46] also reported an annual loss of 2834 hectares of woodland in the western lowlands of Ethiopia for the past 25 years (1985–2010).

A relatively large number of studies have been conducted in the highlands of the central and northern parts of the country, where land degradation and deforestation were already a serious problem many years ago [20,47–50]. These areas experienced either complete removal or massive reduction in forest cover [24,51–56]. For instance, Zeleke and Hurni [24] reported that 99% of the forest cover in the Dembecha area in the northwestern highlands of Ethiopia was converted to agricultural land between 1957 and 1995. Likewise, Amsalu, Stroosnijder [51] reported an 83% reduction in natural vegetation in Beressa watershed in the central highlands of Ethiopian from 1957 to 2000.

Contrary to all these reports and the findings of this report, however, Bewket [57] reported an increase in forest cover for the last four decades in Chemoga Watershed in the Blue Nile Basin of Ethiopia. The increase was ascribed to households' tree planting practices of mainly Eucalyptus species dictated by the growing scarcity of wood for fuel in the area.

### 3.3.2. Patterns of Land Cover Change in the Different Periods of Analysis

The pattern and rate of woodland cover change in the study area varied slightly from place to place and across the different periods considered in this study. Based on the results of the image analysis (Figure 2) and the change analysis made for the five sub-periods (1973–1986, 1986–2000, 2000–2005, 2005–2011, and 2011–2013) (Table 5), it appears that changes in woodland cover did not occur at the same time in all places within the study site. The eastern and southeastern, and the northern part of the study area were the first to experience woodland cover change mainly to agriculture and grassland, respectively. Then, the conversion of woodlands to agriculture and grassland advanced into the remaining parts with the western and southwestern parts being the last destination.

Consequently, the pattern of woodland cover change was explained in terms of the relative suitability of the area for living and crop cultivation as well as access to the road to sell wood and agricultural products. It was explained by key informants and FGD participants that the western and southwestern parts of the study area were considered to be the most hostile environment for living and women who get married to a person living in these areas were considered to be unlucky. According to the same sources of information, the soil and rainfall condition in these areas are considered to be worse than even the adjacent areas, making crop production in particular, and life in general, a challenge.

Additionally, most of these areas are found far from the main road connecting the capital, Addis Ababa, to the southern provinces including Hawassa town, or are rugged in topography, making

transportation of wood products, especially, to market outlets a problem. It is worthwhile to mention here that most charcoal and firewood used to be sold along the roadside. Thus, a relatively better woodland cover is currently found in these areas. Conversely, the less woodland cover is found in areas where the fertility of the soil is relatively good, and sedentary agriculture is intensively practiced and where access to the main road is easy. These views of key informants and FGD participants are consistent with the reports of Eshete [27] and Bekele [58].

Across time scales, most of the land cover changes took place in the first period of analysis, i.e., between 1973 and 1986. This period was also a transitional period, where the socialist (Derg) regime ended the reign of the Imperial regime in 1974. Following the transition, a new land reform proclamation that declares "land for the tiller" was articulated and announced in 1975, which is a radical change in terms of property relations, as land in the previous regime was owned by the state and in the hands of the landlords. All lands including forestlands were nationalized and put under public ownership. The feudal system and landlords were abolished, and the land was distributed to the then tenants and the public through the newly organized peasant associations. By this time, peasant associations in the study area claimed large tracts of woodlands without having clear regulatory provisions as to what rights and duties are vested in the individuals and communities over woodland management and use. This sudden policy and institutional change that destroyed the old without properly entrenching the new turned the woodland into an open-access resource as properly explained by Bekele [58].

According to key informants and FGD members, therefore, this change was one of the main reasons for the massive destruction of woodlands as the local people were in a rush to own and expand agricultural lands. Indeed, it was explained that visible clearing of woodlands for agriculture started earlier than this period when the royal family started to establish and run small-scale modern farms in the late 1950s and 1960s. Bekele [58], Garedew, Sandewall [23], and Biazin [31] also reported similar cases in their study sites.

Moreover, this policy and institutional change were made in the aftermath of one of the most severe droughts that Ethiopia has ever experienced during the turn of the Imperial period (from 1971 to 1973). Hence, the informants added that this drought incidence was another catalyst exacerbating the destruction of woodlands during this period, as the affected people learned to generate income from the sale of charcoal and firewood to buy food and overcome its impacts. According to the same sources, the 1984/85 droughts also had a marked impact, although the people, the poor in particular, had continued selling charcoal and firewood, since they had learned to make them, in order generate additional income for their households even in the absence of drought.

Some studies reported similar findings during the same period (1973–1986), while others reported either generic change trends or changes that were not disaggregated into smaller and similar periods, which in this case makes it difficult to perform a comparative analysis. In the CRV area Muzein [34], Garedew, Sandewall [23], and Biazin [31] reported similar results. Muzein [34] reported that there were more active LULC change processes in the area from 1973 to 1986 than the rest of his study periods. In the same way, Garedew, Sandewall [23] stated that major deforestation and forest degradation took place from 1973 to 1986. He also added that the severe drought of 1984/85 contributed to this change. Biazin [31], on the other hand, reported a massive increase in cultivated land (110%) between 1965 and 1986 at the expense of woodlands. He further added that severe forest loss took place during and in the aftermath of government transitions.

Outside of the CRV area, there are relatively numerous reports with consistent findings. For example, Abate and Lemenih [22] reported that most of the forest cover conversions in the Nadda Asendabo watershed in Southwestern Ethiopia took place between 1973 and 1986. Similarly, Kindu, Schneider [44] reported a reduction in the woodland cover of 81.8%, 52.3%, and 36.1% in the Munnesa area between 1973 and 1986, 1986 and 2000, and 2000 and 2012, respectively, indicating that the major change happened during the 1973 to 1986 period. Shiferaw [59] also reported a major land cover change between 1972 and 1985 in Borena, Northeastern Ethiopia.

On the other hand, Yeshaneh, Wagner [56], with a slight time difference, reported that most of the deforestation in the Koga catchment, northwestern Ethiopia, took place between 1970 and 1980. Likewise, in a study conducted in the Ethiopian central highlands, Beressa watershed, Amsalu, Stroosnijder [51] reported that the major deforestation in the central highlands of Ethiopia was carried out between 1957 and 1984. He also stated that the major land reform that took place in 1975 has led to the clearing of extensive protected forest areas. Dessie and Christiansson [60] too reported that large areas of forest were cleared during periods of political transitions. Congruently, Tegene [54] reported that a significant conversion in Derekolli catchment, northeastern Ethiopia, occurred between 1957 and 1986.

However, contrasting findings have also been reported by Tsegaye, Moe [35], Tefera [19], and Meshesha, Tsunekawa [33]. Tefera [19], although his study period was between 1984 and 2007, reported that the major change in woodland cover took place between 2002 and 2007. The causes of change described include the removal of plants for farmland preparation, fuelwood, construction wood, charcoal, and traditional farm implement making.

Tsegaye, Moe [35], on the other hand, reported a major woodland cover change between 1986 and 2007 (90% reduction), rather than between 1972 and 1986 (67%), in the Northern Afar rangelands. The major reasons given included increased sedentarization of the Afar pastoralists during the period (1986 and 2007) and high influx of migrants from the Tigray highlands, particularly after the 1984/85 droughts, which resulted in huge destruction of woodlands and expansion of cultivation. Similarly, Meshesha, Tsunekawa [33] reported a higher woodland cover loss between 1985 and 2006 (55.6%) than the change observed between 1973 and 1985 (30.7%). The major causes mentioned were population and livestock growth, unsustainable farming techniques, poverty, and the land tenure system. Such contrasting findings signify that land use/land cover changes are influenced not only by policy and institutional changes but also in local socioeconomic contexts.

Although major changes in the study area took place between 1973 and 1986, the conversion of woodlands to other land uses continued persistently throughout the study periods except one (2005 to 2011). In this period (2005–2011) there was an indication of a reversal from agriculture and grassland to woodlands and forests. There was no particular reason given by the informants for the slight temporary increase in woodlands and forests, and reduction in cultivated land and grassland during this period. However, it appears that the government has put more effort into stopping further degradation and protect the woodlands.

The current government, after taking over from the socialist regime in 1991, observed the intensified destruction of woodlands, as did the community, and the gap between policy and practice on the ground. After the takeover, the new government did not reinstitute the necessary tools and organizations quickly enough to stop tree felling and protect forests as explained by the informants and also Bekele [58]. Hence, it was during this period that the government enacted the forest proclamation of the country in 2007, which addresses the development, conservation, and use of forests. The implementation of this proclamation and the associated efforts might have contributed to the temporary reversal of land covers between 2005 and 2011. However, the reversal does not last long and massive woodland and forest areas were converted back to agriculture and grassland after 2011, as observed from the results of the 2013 satellite image (Figure 2) and the figures extracted from the analysis (Table 4).

The most recent analysis period (2011 to 2013) was conducted to detect active changes in land cover and remarkable changes were observed in this short period. The recent satellite image analyzed indicated that deforestation was still actively progressing, and that there was massive destruction of woodlands.

## 4. Conclusions

The results revealed that major land cover changes have occurred over the last forty years. The landscape of the area has dramatically changed from a forested landscape to a predominantly agricultural landscape. Forty years ago, almost all the area was covered by woodlands and forests,

and agriculture was insignificant. Bare land by then was also very small. However, over the last forty years, agriculture and grassland expanded significantly at the expense of woodlands and forests, and forests, which were already small in size, have almost disappeared from the landscape of the study area.

The highest woodland cover change occurred between 1973 and 1986. This period was also characterized by two major incidences in the recent Ethiopian history. It was a transitional period, where the socialist Derg overthrew the longstanding Imperial regime and made fundamental changes, including land tenure rights. It was also a period where Ethiopia experienced two of the most severe droughts in 1971/73 and 1984/85. The woodland decline during this period is, therefore, attributed mainly to these two major encounters in the country. However, the conversion has continued during the other periods as well, except the 2005–2011 period, where there was a temporary gain in the area. The latest analysis period (2011–2013) indicated that woodland conversion is still actively progressing. It should, however, be noted that despite most popular beliefs and some research reports that all is gone, there are still some woodlands, and all is not lost just yet, particularly in remote areas.

Unless action is taken to halt woodland cover change, there is no reason woodlands won't experience the same fate as forests, which have almost disappeared from the landscape of the study area. Urgent attention must, therefore, be paid by relevant stakeholders to arrest woodland degradation and safeguard the fragile ecosystem before it reaches the irreversible stage. Hence, better policy and implementation tools as well as better forest management strategies and practices need to be in place for woodlands to continue providing vital ecosystem goods and services that can improve the livelihood of smallholders in the area as well as the society at large. Additionally, necessary actions and care must be taken during government transitions and drought incidences to avoid similar encounters and reduce woodland loss.

**Author Contributions:** Conceptualization, D.M., B.S., A.B., J.W.R.; Methodology, D.M., H.T.; Original draft, D.M., Editing and writing, D.M., J.W.R., B.S., A.B.; Funding acquisition, A.B., D.M.; Data curation, D.M., A.B. H.T.; Supervision, B.S., J.W.R. All authors have read and agreed to the published version of the manuscript.

**Funding:** This research was financially supported by Department for International Development (DfID) and the Climate Impact Research Capacity and Leadership Enhancement (CIRCLE) program, grant number (201871) and the Volkswagen Foundation project (Ref. No. I/83 735).

**Conflicts of Interest:** The authors declare no conflicts of interest.

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
