# Peer review of "Woodland Cover Change in the Central Rift Valley of Ethiopia"

_forests, doi:10.3390/f11090916_

Round 1

Reviewer 1 Report

The paper presents the issue of land cover change in the central part of Ethiopia, with focus on the woodland areas. The paper proposes an interdisciplinary methodology, albeit insufficiently described.

The paper requires major changes, with restructuring and completion of Introduction, Methodology, Results and Discussions.

Detailed comments are presented below.

Line 26-29 too colloquial

Line 55 Replace “They used to have once covered” with “They once covered”

Line 73-75 Unclear. Please rephrase

Line 131 in table 1 – replace reflective with multispectral

Lines 135-138. Image processing, classification and comparison is insufficiently described. The authors have only mention the ground truth points used for image validation (I assume they mean image classification validation). Maximum likelihood classification is based on spectral signatures extracted from training areas defined on the classified images. There is no mention of the data sources used for the delineation of such training areas on the more recent image, not to mention the previous images.

There is a mention of ArcGIS being used for mapping purposes. What were those purposes?

The authors state that they used supervised classification for land cover classification and change detection analysis. This should be detailed: have they used multitemporal stacks or just classification and subsequent data comparison in GIS software?

Line 140-144 The authors stated that the interviews have been statistically analyzed with SPSS software. Yet no mention is given regarding the structure of the interview, scales used or analysis method. Same for Key informant interviews. These structures should be subjected as supplementary materials.

Line 166 The paper suffers from general poor structuring. The Results should be clearly separated from the discussions and all the methodology aspects that are mentioned in this section transferred above.

Lines 167-190 There are important sections that belong in the methodology (class description, classification details). No mention of the validation method was done in methodology (accuracy, kappa statistics). Comparisons with literature should be transferred to discussions.

Line 174 The authors use the terms land use and land cover interchangeably in this section and throughout the paper. In the column title, the term used is land cover type, yet in the description land use features are presented (type of management undertaken to preserve said land cover). The terms need to be used according to definitions in the literature.

Line 224 Again, the results of the study need to be clearly separated from the discussions. Is not clear which sections refer to original results and which are taken from literature.

The authors mention in the methodology section a change detection analysis. The results presented show only area distribution on land cover at different times, without any from-to change detection, which could easily be obtained from classified image overlay. These changes could also be presented graphically, with maps of land cover change, which would illustrate the pattern of change in different areas. Especially since in lines 380-390, the authors refer to reversible changes of the forest and woodland areas. The RADAR Chart is not conclusive change detection analysis, since it only shows changes in the total areas per classes. A change detection matrix per each interval would be far more conclusive.

In section 3.3.2, there is information regarding the results of FGD and key informants, mixed with site description, information on institutional and legislative changes and discussions of the results. To solve these issues, the authors need to:

  • Include a section in the introduction to describe in more detail the institutional and legislative framework of the area during the time period, as it is described to
  • Present the results of the analysis, separately on the stages presented in the methodology: change detection, household survey, FGD discussions, Key Informant interviews
  • Literature comparison done in the discussions section, to cross-reference the different types of results with previous studies

Line 280-288 The authors mention conversions between different land cover types based on the classified images and tables, eve though there is no pixel-based change detection analysis was properly done

The social aspects of the research focus only on reasons for deforestation and do not tackle the issue of farmers’ awareness of the problems related to land use change, since the main driver of the process is represented by extensive farming (from what is understood from the paper). I would have appreciated a presentation of such results, in case the interviews contained this line of questions.

Line 414-416 Unclear. Please rephrase

Reviewer 2 Report

This manuscript (ms) described a study on land cover change in the Central Rift Valley (CRV) of Ethiopia. The background and related works are well introduced. The experiment is well organized. The language is good. The author should give more information about the methodology. Some detailed issues need to be addressed. I think the ms needs moderate revisions to be acceptable.

General remarks: 

The ms has given a broader view of land cover land use (LULC). But the introduction section should also include a brief review of methodology (e.g., machine learning) in the LULC study. I suggest the authors add a paragraph to briefly introduce some state-of-the-art methods and tools that are widely adopted by the LULC community. Here are some good references:

Sidhu, N., Pebesma, E., Câmara, G., 2018. Using Google Earth Engine to detect land cover change: Singapore as a use case. European Journal of Remote Sensing 51, 486–500. https://doi.org/10.1080/22797254.2018.1451782

Zhang, C., Di, L., Lin, L., Guo, L., 2019. Machine-learned prediction of annual crop planting in the U.S. Corn Belt based on historical crop planting maps. Computers and Electronics in Agriculture 166, 104989. https://doi.org/10.1016/j.compag.2019.104989

Halmy, M.W.A., Gessler, P.E., Hicke, J.A., Salem, B.B., 2015. Land use/land cover change detection and prediction in the north-western coastal desert of Egypt using Markov-CA. Applied Geography 63, 101–112. https://doi.org/10.1016/j.apgeog.2015.06.015

The land cover maps were produced using pixel-based supervised image classification with a maximum likelihood classification algorithm. The result looks good but the detailed information about the classification process is lacking. There is no methodology section throughout the ms. Which bands of Landsat data were used in classification? Was there any cloud coverage issue? The author should add a section to systematically describe the classification method. 

The authors used a lot of survey data. These data were used to validate the classification result. So what do these data look like? How is the accuracy other than 2013? The authors should find a way to present the survey data, preferably a map, so readers can better compare the land cover map result with ground truth. 

Figure 2 tries to display the land cover change “between 1973 and 2013”. Since the experiment includes six land cover maps, the change over time (i.e., 1986, 2000, 2005, 2011) should be reflected in the chart as well. The authors may consider presenting the change for each land cover class in trend charts. 

Round 2

Reviewer 1 Report

The authors have addressed the comments in the review and have restructured most of the paper to meet the requests.

Therefore, I agree with the publication of the paper.

Reviewer 2 Report

The point-to-point response to my comments and suggestion was adequate and thorough. Thank you for the revisions.